psychology

visual complexity, attention, perceptual load

**Author for correspondence:**
Fintan Nagle
e-mail: fintan.nagle@ucl.ac.uk

# Predicting human complexity perception of real-world scenes

## Fintan Nagle and Nilli Lavie

Institute of Cognitive Neuroscience, University College London, London, UK

 FN, 0000-0003-0514-4356; NL, 0000-0001-5274-0535

Perceptual load is a well-established determinant of attentional engagement in a task. So far, perceptual load has typically been manipulated by increasing either the number of task-relevant items or the perceptual processing demand (e.g. conjunction versus feature tasks). The tasks used often involved rather simple visual displays (e.g. letters or single objects). How can perceptual load be operationalized for richer, real-world images? A promising proxy is the visual complexity of an image. However, current predictive models for visual complexity have limited applicability to diverse real-world images. Here we modelled visual complexity using a deep convolutional neural network (CNN) trained to learn perceived ratings of visual complexity. We presented 53 observers with 4000 images from the PASCAL VOC dataset, obtaining 75 020 2-alternative forced choice paired comparisons across observers. Image visual complexity scores were obtained using the TrueSkill algorithm. A CNN with weights pre-trained on an object recognition task predicted complexity ratings with $r = 0.83$. By contrast, feature-based models used in the literature, working on image statistics such as entropy, edge density and JPEG compression ratio, only achieved $r = 0.70$. Thus, our model offers a promising method to quantify the perceptual load of real-world scenes through visual complexity.

## 1. Introduction

Research on the role of attention in perception has emphasized the role of perceptual load in an attended task as a determinant of the level of attentional engagement [1–4] and conversely of inattentional blindness to unattended stimuli [5–7]. Specifically, when compared to low load tasks, conditions of high perceptual load have been shown to result in higher attentional engagement in the task as well as reduced perception of, and brain response to, unattended stimuli [8–12]. High load is usually operationalized by using visual search tasks with a higher number of items or a greater proportion of target-similar items, by using a more

heterogenous distractor array, or by using more complex processing requirements (such as feature conjunctions) which do not allow pop-out [1–10,12–15].

To date, most of the load manipulations used have typically involved rather simple stimuli, for example letters, shapes, and a few objects; see [13–16] and for reviews [1–4]. It therefore remains unclear how perceptual load can be operationalized in real-world stimuli such as images of natural scenes. Accordingly, previous research on the role of attention in natural scenes has not as yet varied the level of perceptual load of the image itself. In a divided attention paradigm, the load of an added primary task in which perceptual load can be more easily manipulated (such as visual search for letters or digits) has been varied while observers performed a secondary natural scene perception task [17]. Studies have also compared attended and ignored images [18]. Thus, the question of how to estimate and manipulate the perceptual load of diverse, real-world images is still open.

Here we aim to predict perceived image complexity in order to use it as a potential proxy for perceptual load in natural scene perception. We collected 75 020 pairwise 2-alternative forced choice (2AFC) complexity judgements ('which image is more complex?') over a set of 4000 natural scene images from the PASCAL VOC dataset [19], widely used in computer vision research on object recognition. From these comparisons, we generated a ranking over the images, assigning a complexity score to each one. We then applied a deep learning approach, training a convolutional neural network (CNN) to predict the complexity scores of these varied real-world scenes.

Previous work on prediction of perceived image complexity, which we review next, has used simple trainable models (shallow neural networks or linear combinations) drawing on hard-coded (non-learnable) low-level features, such as the edge density or orientation variance; this approach has resulted in limited predictive power.

## 1.1. Complexity and visual clutter

The concept of visual complexity is related to that of visual clutter, which Rosenholtz et al. [20] define as 'the state in which excess items, or their representation or organization, lead to a degradation of performance at some task.' This work also introduced the influential feature congestion model, which computes local image statistics such as colour variance. It is based on hand-crafted features rather than learned parameters.

This model has been frequently extended. Deza & Eckstein [21], for example, presented an adaptation of the Rosenholtz feature congestion model to a foveated context, using increased resolution at the image centre. This model was validated against visual search reaction time (RT) judgements.

Yu et al. [22] proposed a proto-object model of visual clutter. They note that good progress has been made at quantifying clutter by counting objects, giving an estimate of set size, and observe that image segmentations are imperfect ground truths for clutter because they are subjective. The proto-object model, thus, provides a middle ground between low-level features and high-level objects. Here, images were segmented into groups of similar pixels using two methods: SLIC superpixels [23] and the entropy rate superpixel [24]. These were then clustered (in colour space) into proto-objects, and clutter was predicted from proto-object count.

To validate this model, observers also ranked a set of 90 images in order of perceived clutter. The model predicted the ranks quite well, achieving Spearman's $\rho = .81$ (as ratings only made sense in terms of rank, Pearson's $r$ was not given).

Is clutter the same concept as complexity? While the concept of 'excess items' appears related to that of complexity, it is less clear that the disorderly organization that reflects higher clutter always indicates higher complexity. One can imagine a scene of high complexity which, owing to an ordered arrangement of objects, does not appear cluttered.

Visual clutter has been studied in close relationship with the psychophysics of visual search and with examples of degradation in object recognition or visual search, such as crowding. Measures of visual clutter are therefore based primarily on performance drops in visual search, usually with simple, well-segmented targets and distractors.

By contrast, our image comparison task attempted to measure the perceived visual complexity of diverse real-world images. Our complexity ratings were not obtained by accuracy or RT measurements of recognition or search. They did not correlate well with visual search RTs on the same images (see §3.7). Our complexity ratings, therefore, are informed primarily by observers' idea of visual complexity as a descriptor.

We note that 'clutter' is more generally used in natural language to denote objects (e.g. 'a cluttered room') and is not as applicable to natural scenes. Object count (set size) is often used as a proxy for clutter. 'Complex' and 'simple', however, apply naturally to any visual stimulus without connoting a profusion of objects. Two scenes with an equal number of objects could be rated very differently in complexity owing to differences in the nature of the objects, their affordances, their textures, and the nature and texture of the background.

If clutter cannot predict complexity, does this suggest that the two concepts differ? We address this question by correlating our complexity ratings with statistics from the Rosenholtz feature congestion model. We also investigate the relationship between complexity and object perception by correlating our complexity ratings with object counts and region counts obtained by image processing and machine learning.

## 1.2. Feature-based models of complexity in natural scenes

Oliva *et al.* [25] were among the first to investigate the perceived complexity of real-world images (in this case, of indoor scenes). They asked observers to split indoor scenes iteratively into groups of high and low complexity, then applied multidimensional scaling (MDS) to embed the scenes into a two-dimensional space. MDS often delivers axes of variance which track image properties such as brightness or scene 'openness'. In this case, apart from a first axis which appeared to correspond to 'clutter', the dimensions suggested did not match clearly to any image features. Although this is not a predictive model, it highlights the difficulty of decomposing complexity into easily-describable features.

Cavalcante *et al.* [26], using photographs of streetscapes, asked participants to rate the complexity of 74 unaltered images. Applying a model using learned independent component filters, they achieved a correlation of $r = 0.72$ with perceived ratings. However, accuracy was not evaluated on a held-back validation set, so the model's predictive power is unclear.

Machado *et al.* [27] developed a predictive machine learning model whose training data was obtained by asking 30 participants to rate the complexity of 800 images using a Likert scale ranging from 1 to 5. The majority of the images were art (abstract or representational art and clip art), but 200 photographs of natural scenes were also included. For each image, 329 features were computed; a feed-forward neural network with one hidden layer was then trained to predict perceived complexity ratings from these features, achieving a Spearman's $\rho$ rank correlation of 0.83 on an unseen validation set.

However, given the mixed classes of images, it is possible that the model learned inter-class differences in complexity ratings rather than the features which underlie complexity in natural scenes across different classes. This is especially possible given that only a quarter of the training set was made up of natural scene images; it thus remains unclear whether this model could successfully predict complexity on a wide range of unseen natural scene images.

Corchs *et al.* [28] trained a machine learning model on a dataset consisting exclusively of outdoor photographs. They asked 26 observers to judge the complexity of 49 real-world scenes using a slider ranging from 0 to 100. Eleven image statistics were extracted, including a measure of 'visual clutter' using the Rosenholtz feature congestion model [20]; a linear combination of these features was fit to the mean complexity ratings using particle swarm optimization (PSO) [29].

Validation was performed by using the same linear combination of features to predict the complexity scores from a second unseen set of 49 high-quality professional photographs of real-world scenes from the LIVE and IVL databases, on which its correlation with perceived scores reached $r = 0.81$. The features with most influence on the final score were the number of regions according to segmentation by the mean shift algorithm, the frequency factor (the ratio between the spatial frequency under which lies 99% of the image's energy, and the Nyquist frequency), and an estimate of the number of colours in CIELAB space. The small size of the the training set (49 images) raises the possibility that the model's applicability to diverse real-world scenes could be low. The validation data, consisting also of 49 images, showed a drop in $r$ of 0.05 (5.8 percentage points), which may be an indication that the model was overfit to the training data.

## 1.3. Our approach

For data collection, most of the studies just described used Likert scales. Here we evaluated perceived complexity using 2AFC paired comparisons; observers were presented with pairs of images and asked to indicate which one was more complex. The 2AFC approach has several advantages over Likert-like ratings. Firstly, it is less subject to response bias. For example, a conservative observer may rank all images lower than a non-conservative observer, but this would not affect their 2AFC choice, which is based on a relative within-pair complexity judgement. Secondly, the method of 2AFC paired comparisons is more resistant to changes in criterion over the course of exposure to more images in the experiment, because observers are forced to choose one image from each pair rather than relating each image to the increasing number of previously rated images.

We therefore generated complexity ratings for 4000 images by presenting pairs of images and requesting observers to make a 2AFC judgement on each pair. In this way, we collected 75 020 comparisons which we then converted into a complexity score for each image.

The studies just described all used the approach of extracting a large number of different features, then using a learning algorithm (simple neural network, support vector machine or other optimizer) to predict complexity ratings. Can these results be improved by using a CNN approach with learned features, and can complexity ratings be predicted across a more diverse set of real-world images? Here we trained a deep learning network to predict the perceptual complexity of 4000 images of a variety of real-world scenes, aiming to achieve a higher level of prediction while avoiding overfitting.

# 2. Experiment 1: complexity of whole images

Using a CNN, we aimed to predict the complexity of each image, rather than the outcomes of individual 2AFC comparisons or the full 4000-by-4000 distance matrix.

There are many approaches to generating a ranking from paired comparisons, going back to Thurstone [30]. Game theoretic techniques, which model comparisons as competitions between images, are commonly used to assign a score to each competitor; examples are the Elo rating system [31] and its extension the Glicko system [32]. We used TrueSkill, a state-of-the-art method which also generalizes the Elo method and is newer than the Glicko method.

## 2.1. Material and methods

### 2.1.1. Participants

Sixty-two observers (11 male) with a mean age of 24.9 (s.d. 9.0) were recruited from the University College London (UCL) Institute of Cognitive Neuroscience subject pool. They all had normal or corrected-to-normal vision and were asked to have a good night's sleep before attending the experiment.

### 2.1.2. Dataset

We used the PASCAL VOC object recognition dataset [19], which consists of real-world scenes containing one or more objects with at least one from the following 20 classes: person, bird, cat, cow, dog, horse, sheep (animals); aeroplane, bicycle, boat, bus, car, motorbike, train (vehicles); bottle, chair, dining table, potted plant, sofa, TV/monitor (indoor objects). The dataset contains images with various aspect ratios taken by different cameras. We used a randomly selected subset of 4000 images.

### 2.1.3. Procedure

We asked observers to choose, from each pair, the image that they thought was most visually complex. Written instructions indicated that 'an image which is more visually complex will take up more of your attention as you look at it.' All observers were given the same instructions at the beginning of the session.

On each trial, observers were shown a green dot to indicate that the trial was ready. They could then hold down the A key to view the first image, or the K key to view the second image. Each image was only displayed when the corresponding key was held down, and only one image was displayed at a time. The number of times each image was viewed was not constrained; nor was the total duration of the trial. We collected the pattern of image viewings, with timings, for each trial. Participants then pressed the Z key if they wished to indicate that the first image was more complex, or the M key to select the second image.

The experiment was deployed using an interactive website built with the Flask server [33] and hosted using Amazon Web Services Elastic Beanstalk. Data collection took place in a UCL computer room and was performed in 90 min sessions each hosting 20–30 observers, who completed varying numbers of comparisons. We presented 12 practice trials at the beginning of the experiment. Data collection was not divided into blocks, and we allowed free breaks. Pairwise comparisons were randomly distributed across participants, so that image presentation counts were approximately equal and no participant was more or less likely to see any particular image. Owing to the croudsourced nature of this experiment, counts were approximately rather than perfectly balanced.

### 2.1.4. Post-processing

The web interface provided us with a list of comparisons, each one effectively a competition in complexity between two images. Each image participated in approximately 37 comparisons. These comparisons were used as input to the TrueSkill algorithm [34], which uses a Bayesian framework to assign a score distribution to each image based on its performance in each competition. We used the

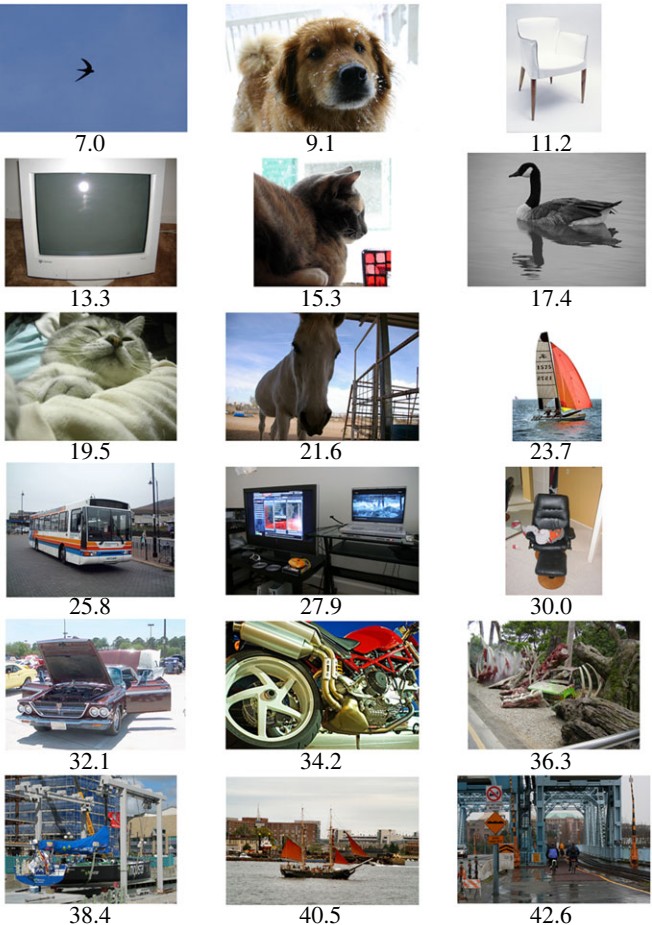

**Figure 1.** Sample images from the PASCAL VOC dataset, shown in order of complexity rating, including the images of lowest and highest complexity. Observers performed 75 020 pairwise comparisons; during each one, we asked 'which image is most visually complex?' Images were then ordered using the TrueSkill algorithm, which assigned a unique complexity score to each image.

mean of this distribution as that image's complexity rating. Individual observer results were not modelled and no competitions between observers were set up. Example images are shown in figure 1.

## 2.2. Results

We collected 75 020 pairwise comparisons of 4000 images. Each of the 62 participants judged on average 1210 comparisons. TrueSkill ratings formed an approximately normal distribution with mean 25.0 and standard deviation 5.5 (figure 2).

### 2.2.1. Consistency of results

We examined each of the comparisons to check whether its result (image A or B) matched the TrueSkill ratings (rating A > rating B if the observer chose A as most complex, and rating B > rating A if B was chosen).

The 2AFC judgements matched the TrueSkill ratings in 76.1% of comparisons, showing that there is sufficient consistency in human ratings for TrueSkill to form an effective model and indicating that it is appropriate to represent perceived complexity as a one-dimensional space.

## 3. Modelling

To model perceived complexity, we trained a CNN to predict complexity ratings directly from pixel-level image data. We examined two CNN architectures and investigated the effects of pre-training. We then measured the predictive power of individual image statistics and pixel-level regression, before comparing to the model of Corchs *et al.*

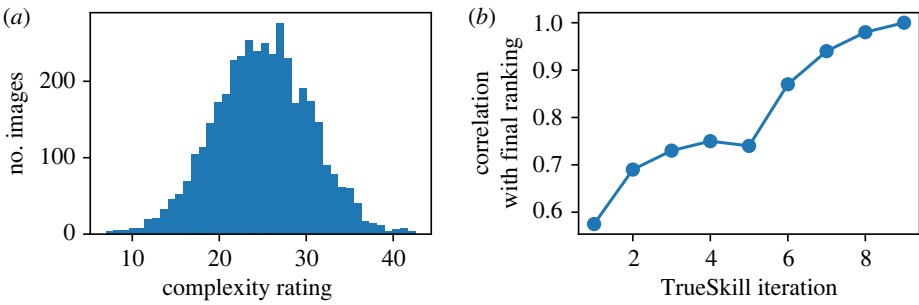

**Figure 2.** (*a*) Final distribution of complexity ratings for all 4000 images. (*b*) Convergence of the TrueSkill algorithm after rating of the first 2000 images (iterations 1–5) and after the addition of the second 2000 images (iterations 6–9). Decreasing slopes show the algorithm's convergence in each case.

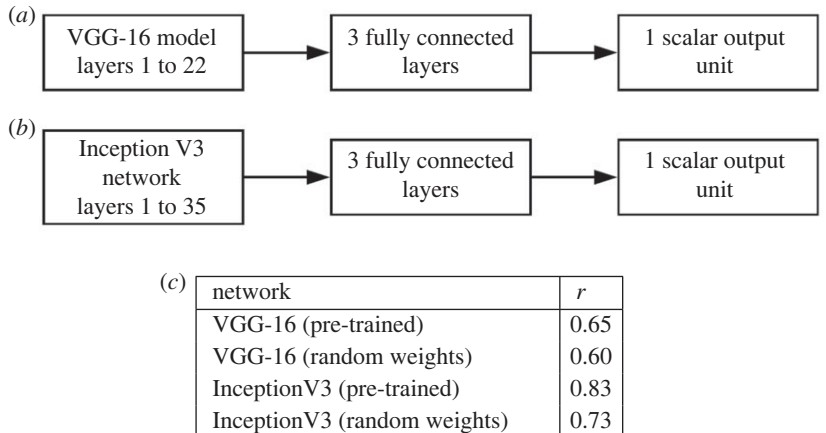

| network | $r$ |
|---|---|
| VGG-16 (pre-trained) | 0.65 |
| VGG-16 (random weights) | 0.60 |
| InceptionV3 (pre-trained) | 0.83 |
| InceptionV3 (random weights) | 0.73 |

**Figure 3.** The networks used to model perceived complexity directly from image data. (*a*) Modified VGG-16 architecture. (*b*) Modified Inception V3 architecture. (*c*) Correlations between perceived complexity ratings and the predictions of four models (evaluated on a 10% validation set unseen during training).

## 3.1. Correlation metrics

We used the TrueSkill algorithm [34], designed for assessing skill in competitive games. An individual's skill is represented as a normal distribution; each individual is tagged with a $\mu$ coefficient, representing skill (distribution mean), and a $\sigma$ coefficient, representing the algorithm's confidence (distribution variance). In our application, $\mu$ represents perceived complexity. TrueSkill converged on a stable ranking, indicating that we had collected enough comparisons.

We estimate our models' predictive power by correlating their complexity estimations with images' TrueSkill ratings. Much of the literature on complexity compares model predictions to perceived ratings using Spearman's rank correlation coefficient ($\rho$), which does not take into account direct values but only their ranks. Because Trueskill's output is a computation generating each image's probability of succeeding in paired comparisons, rather than a Likert scale, we instead used Pearson's $r$ correlation coefficient as it is sensitive to value as well as rank.

## 3.2. Deep learning

We used modified versions of two architectures, VGG-16 and Inception V3 (figure 3) to model complexity. Both networks were originally trained on the ILSVRC object recognition dataset [35], which contains approximately 1.2 million images in 100 labelled classes. Images were sampled from the larger ImageNet dataset [36], which contains approximately 50 million images. Here, both networks were evaluated using the pre-trained weights from ILSVRC data, as well as being trained from scratch with randomly initialized weights.

For the VGG-16 architecture, we removed the final softmax (discrete classification) layer, replacing it with six fully-connected layers with {512, 256, 128, 64, 32, 1} units and terminating in a rectified linear unit (ReLU) outputting a scalar complexity estimate. Neither inputs nor outputs were scaled; as is

standard practice, mean input pixel values were shifted closer to zero by subtracting the means of the ILSVRC colour channels (103.939, 116.779 and 123.680 for red, blue and green respectively). For the Inception V3 network [37], we removed the final classification layers and replaced them with five fully-connected layers with {1024, 512, 64, 32, 1} units, the final layer outputting a scalar complexity estimate.

Accuracies were calculated on a randomly selected held-back validation set consisting of 10% of the dataset (400 images), which was unseen by the network during training. Results are shown in figure 3. The pre-trained Inception V3 network achieved the highest correlation with perceived complexity ratings, $r = 0.83$. In both cases, there was an accuracy drop when pre-trained weights were replaced by randomly initialized weights.

### 3.2.1. Linear regression on raw pixel vectors

To provide a baseline for pixel-level learning, we performed linear regression on raw pixel vectors. Each colour image was converted into a vector whose length was height × width × 3, corresponding to the red, green and blue (RGB) intensities of each pixel. We used 10-fold cross-validation, holding back 10% of the images as a validation set on each iteration and submitting the rest for training. This linear regression model achieved a correlation of $r = 0.62$ with perceived complexity ratings.

## 3.3. Correlations between complexity and image features

To assess the contribution of image features (from low-level properties such as entropy or edge count, to higher-level features such as estimated salience), we evaluated the correlation between 38 basic visual features (including 12 used by Corchs *et al.*) and perceived complexity ratings. These were all scalar features, with each one assigning a single number to each image.

Several features were used in the model of Corchs *et al.* Four of these features were calculated from the grey-level co-occurrence matrix using MATLAB's `graycoprops` function. Colourfulness is a linear combination of an image's mean and standard deviation in CIELAB space [38]. Colour count and colour harmony are calculated by software described in [39].

Other features are due to established models. We obtained the variance of each of the RGB channels, as well as the variance of the flattened image matrix. We calculated the entropy of the RGB and hue, saturation, value channels, as well as that of the whole image. Colour clutter (and its mean and variance) were calculated using the feature congestion model due to Rosenholtz *et al.* [20]. We calculated mean luminance by transforming images to monochrome and averaging pixel values. Image size was taken as the number of pixels in an image. Edge density was calculated using the Canny method [40] with threshold [0.11, 0.27] and $\sigma = 1$. JPEG compression ratio was obtained by dividing the number of bytes in an image's uncompressed representation by the size in bytes of its compressed version. The mean shift region count was found using the Matlab wrapper [41] to the EDISON implementation [42] of the mean shift algorithm [43].

To investigate the effect of mid-level features, we extracted local keypoints from each image using the scale-invariant feature transform (SIFT) [44] and speeded-up robust features (SURF) [45] algorithms. We found between 7 and 4217 SIFT keypoints (mean 807) and between 12 and 4355 SURF keypoints (mean 1472).

We estimated segment counts using the maximally stable extremal regions (MSER) [46] algorithm, which found between 0 and 3382 regions (mean 360). We also used the deep Mask R-CNN model [47] to detect and thus count objects in images.

We generated salience maps using the MLNet [48] and SalGAN [49] deep models, then calculated their mean and their entropy. Finally, visual search RTs come from human data obtained from Ionescu *et al.* [50]; observers were asked to search for one of the objects tagged in the PASCAL VOC dataset.

None of the measures showed $r > 0.5$ when correlated individually with complexity ratings; the most informative was the Mask R-CNN object count ($r = 0.49$). Results are shown in table 1.

## 3.4. Predicting complexity from scalar image features

We tested the usefulness of the feature combination approach by using these 38 features to train three models: a linear regression, a support vector machine, and a 3-layer feed-forward neural network. To deal with nonhomogeneity of variance, inputs were all rescaled to zero mean and unit variance. Complexity ratings were not rescaled.

Straightforward linear regressions, evaluated on an unseen, randomly selected 10% validation set, achieved on average $r = 0.65$. Because the validation set was always unseen during training, this model's accuracy was not inflated by overfitting. We also conducted a full leave-one-out cross-

**Table 1.** The 38 image statistics and features which we evaluated as predictors of complexity ratings. (This table shows their correlations ($r$) with perceived complexity ratings as well as their mean linear regression coefficients ($b$), averaged over 4000 runs of leave-one-out cross-validation.)

| category | statistic | $r$ | $b$ |
|---|---|---|---|
| features from the grey-level co-occurrence matrix | contrast | 0.30 | 0.32 |
| | correlation | −0.11 | 0.01 |
| | energy | −0.35 | −0.14 |
| | homogeneity | −0.28 | 0.78 |
| colour-related features | colourfulness | 0.13 | −0.18 |
| | colour count | 0.47 | 0.82 |
| | colour harmony | −0.26 | −0.12 |
| colour channel entropy | R | 0.36 | 0.45 |
| | G | 0.34 | −0.1 |
| | B | 0.30 | −0.23 |
| | H | 0.27 | 0.1 |
| | S | 0.26 | −0.13 |
| | V | 0.39 | 0.02 |
| variance | all | 0.16 | 0.54 |
| | R | 0.18 | −0.17 |
| | G | 0.17 | −0.43 |
| | B | 0.17 | −0.03 |
| feature congestion model | contrast clutter: mean | 0.38 | 0.27 |
| | contrast clutter: variance | 0.31 | 0.25 |
| | colour clutter | 0.35 | −0.52 |
| | sub-band entropy | 0.05 | −0.13 |
| salience models | mean ML-net salience | 0.17 | −0.02 |
| | ML-net salience map entropy | 0.22 | −0.11 |
| | mean SalGAN salience | 0.27 | 0.14 |
| | SalGAN salience map entropy | 0.30 | 0.22 |
| mid-level features | SIFT count | 0.33 | 0.25 |
| | SURF count | 0.40 | 0.32 |
| proto-object features | mean shift region count | 0.36 | −0.22 |
| | MSER count | 0.45 | 0.32 |
| object features | Mask R-CNN object count | 0.49 | 0.77 |
| other | visual search reaction time | 0.37 | 0.46 |
| | mean luminance | −0.01 | −0.02 |
| | image size (pixel count) | 0.00 | 0.01 |
| | whole-image entropy | 0.43 | −0.26 |
| | frequency factor | 0.18 | 0.22 |
| | edge density | 0.42 | −0.24 |
| | JPEG compression ratio | −0.27 | −0.08 |

validation, obtaining $r = 0.70$ over 4000 iterations. The observed accuracy gain of 5 percentage points shows that a larger training set led to more accurate predictions.

To further rule out overfitting effects, we trained models with lasso (L1) and ridge (L2) regularization. In each case, we trained the model on 90% of the data, performing a grid search across 10 values of $\alpha$

**Table 2.** Correlations with perceived complexity given by models fit to vectors of image statistics and features.

| model | r |
|---|---|
| linear regression (4000 runs, leave-one-out cross-validation) | 0.70 |
| lasso regression (10 runs) | 0.69 |
| ridge regression (10 runs) | 0.69 |
| support vector regression (10 runs) | 0.60 |
| neural net (100 runs) | 0.68 |

from $10^{-15}$ to 20 and converging on $\alpha = 0.0001$ (lasso regression) and $\alpha = 0.01$ (ridge regression). This process was repeated 10 times. Mean accuracy on the held-back validation sets was $r = 0.693$ for lasso regression and $r = 0.694$ for ridge regression.

A support vector regression (SVR) achieved $r = 0.60$ on unseen validation sets using 10-fold cross-validation. We also trained a 4-layer feed-forward neural network with 38 inputs and {128, 64, 32, 1} units. ReLU activation was used throughout. We performed 10-fold cross-validation; the mean correlation with complexity ratings was $r = 0.68$.

These results (table 2) show convincingly that it is possible to predict real-world scene complexity from basic and mid-level image features at approximately $r = 0.70$. All accuracies were evaluated on unseen training data and a full leave-one-out cross-validation showed the highest correlation with human complexity ratings. The regression coefficients for each image feature, averaged over 4000 runs of leave-one-out cross-validation, are reported in table 1. Predictions were thus improved by incorporating multiple features, but were still of low accuracy compared to the CNN model.

## 3.5. Predicting complexity by characterizing mid-level features

Mid-level features such as SIFT and SURF aim to characterize the local neighbourhoods of keypoints in an image. We have shown that counting extracted keypoints gives some information about an image's complexity. Are the values of these descriptors also useful?

To investigate, we generated SIFT and SURF descriptions of constant length by sampling 500 keypoints from each image, with replacement. This allowed us to analyse the character of the keypoints while controlling for their number (which is treated by the feature-based models). We assembled each image's keypoints into a vector of length $500 \times 128 = 64\,000$ (SIFT) or $500 \times 64 = 32\,000$ (SURF).

Image vectors were then used to train an SVR on a randomly selected 90% training set. We were not able to find parameters enabling the SVR to make effective predictions, so we do not report these results.

However, a neural network was able to predict complexity values from sampled SIFT and SURF descriptors. We trained a 4-layer feed-forward neural network with 38 inputs and {128, 64, 32, 1} units. ReLU activation was used throughout. We performed 10-fold cross-validation. The mean correlation with complexity ratings was $r = 0.45$, showing that SIFT and SURF descriptors can inform on perceived complexity, but are not the best predictors. Further work could use an recurrent neural network instead of a feed-forward neural network, allowing all keypoints from images with different numbers of keypoints to be taken into account.

### 3.5.1. Comparison to the model of Corchs *et al.*

We replicated the model of Corchs *et al.* [28] by using the same features and regression coefficients. We then applied it to our images and correlated its predictions with perceived ratings; Pearson's $r$ was 0.47, showing that learning did not transfer well to our dataset from the 49 images on which this model was trained. This drop in performance shows that this model is not applicable to a wide range of natural and artificial scenes.

## 3.6. Object recognition and complexity

Here we investigated the influence of approximate object counts and object class information on complexity perception.

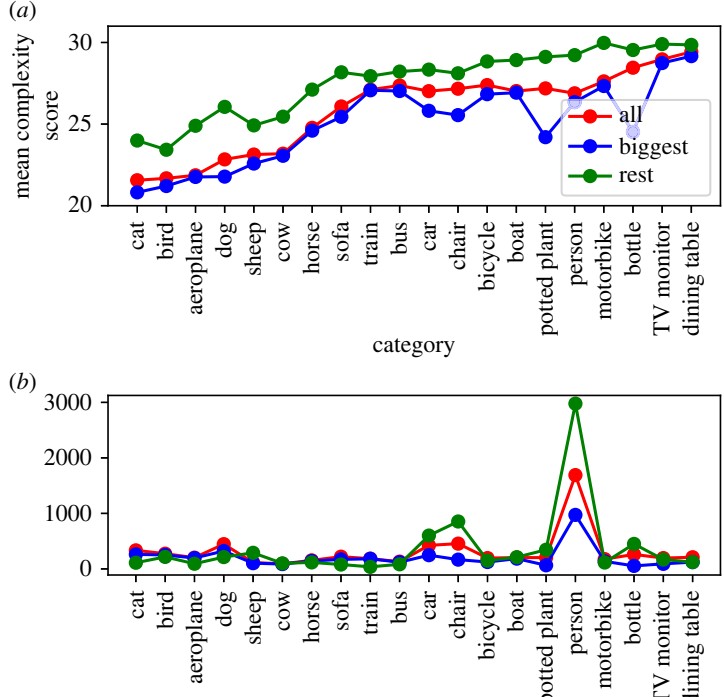

**Figure 4.** Object classes and their effect on complexity ratings. (*a*) Object classes ordered by mean complexity score. The top line (green) shows the mean score of images containing this class as objects which are not the largest. The bottom line (blue) shows the mean score of images containing this class as the largest object. The middle line (red) shows the mean score of images containing this class anywhere in the image. (*b*) The number of images containing each object, which is relatively constant with the exceptions of cars, chairs and people. Colours are as in (*a*).

### 3.6.1. Number of objects

How does the amount of objects present in an image influence its complexity perception? To investigate, we estimated the number of objects in each image. Because human observers do not always agree on this, we used the Mask R-CNN architecture [47] to detect and count multiple objects. Images were processed with an implementation trained on the MS COCO dataset [51] containing 98 object types. While this network does not detect all objects, it allowed us to generate an unsupervised estimate of object count.

Object count and complexity rating were correlated at $r = 0.49$. The next most highly correlated visual feature was the colour count, at $r = 0.47$; object count is thus the most useful individual predictor of complexity.

### 3.6.2. Object categories

Here we asked whether the presence of particular object categories can be predictive of an image's complexity rating. Such an effect could occur owing to a form of cueing based on inherent associations between some object categories and the nature of the scene; for example, an image containing a sheep is likely to be set in a less busy environment than one containing a taxi.

To investigate, we looked at the mean complexity ratings of images containing each category. PASCAL VOC images contain multiple annotated objects (min: 1, max: 56, mean: 2.8), each with a bounding box. We first calculated the mean complexity ratings of all images containing an object belonging to a particular category.

Many of the objects present are very small, and images often contain one object which is much larger than the others. To investigate the influence of these larger objects, we isolated in each image the object with the largest bounding box, referring to it as the 'central object' and to the rest as 'background objects'. We then re-calculated the mean complexity scores, considering an image to contain a particular category only if that object was present as a central object. Finally, we looked at the mean complexity scores for groups of images containing a particular category, but not as the central object (i.e. in the background).

**Table 3.** Correlations with perceived complexity ratings of predictions from the PASCAL VOC object taggings. (We predict complexity as the mean rating of images containing the biggest object; the sum of the mean complexities for image sets containing each object class in the current image; the mean of the same; the sum of the mean complexities for image sets containing each object class in the background of the current image; the mean of the same; the number of tagged objects in the image; a support vector regression (SVR) on the object presence vectors (0 for absent, 1 for present); and an SVR on the object count vectors. SVR accuracy was evaluated on a 10% held-back validation set.)

| model | r |
| --- | --- |
| biggest object | 0.42 |
| all objects (sum) | 0.42 |
| all objects (mean) | 0.45 |
| background objects only (sum) | 0.38 |
| background objects only (mean) | 0.43 |
| number of tagged objects | 0.40 |
| SVR output from object presence vectors | 0.49 |
| SVR output from object count vectors | 0.56 |

Figure 4 shows the mean complexity scores by object class, along with counts for the images in each set. For each object class, the set of images with that class in the background had higher mean complexity than the set of images with that class as a central object; the set of images with that class present anywhere has a mean complexity score between those two values (because it is their mean). Object classes thus appear in slightly higher-complexity images when they are in the background. This may be owing to an object in the background having a greater likelihood of appearing in conjunction with other objects, increasing scene complexity.

It is possible that there are also some effects reflecting photographers' framing tendencies. For example, images containing cats may often be less complex owing to a tendency to depict cats alone and centrally in the frame, and dining tables are perhaps more likely to appear in conjunction with multiple other objects. However, the large distributions of complexity scores within object classes show that object class does not fully determine complexity.

To further assess whether the object class information in the PASCAL VOC taggings is informative for complexity prediction, we predicted each image's complexity in various ways from the object taggings, then correlated the results with perceived complexity ratings. For each object class, we calculated the mean complexity of images containing that class and used it directly to predict individual image complexities. Results are shown in table 3; mean class complexities were better correlated with perceived ratings than summed class complexities, suggesting an integrative rather than an additive process. The best-performing model was an SVR trained on object count vectors.

## 3.7. Complexity and visual search reaction time

We compared complexity ratings to human visual search RTs obtained by Ionescu *et al.* [50]; there was a fairly low correlation between the two sets of scores ($r = 0.37$). This is not surprising because the two tasks (judgements of complexity and search for a particular object) were very different. Correlations of this size are typically found among different tasks which share a common cognitive factor, specifically one that relates to the perceptual processing demands of the task [52,53].

Moreover, in busy real-world scenes, visual search RT is expected to depend not only on image complexity but also on other factors that are specific to the relationship between the search target and the scene characteristics: salience, for example. It is easy to imagine a highly complex image with a very short visual search time (searching for a bright red balloon in a busy grey cityscape), as well as an image of low complexity with a long visual search time (searching for a snowy owl against a snowy background). Semantic contextual factors can play a role too: for example, the RT to detect that a sofa is absent should be lower in an image of an outdoor scene than in an image of an indoors scene irrespective of their level of complexity.

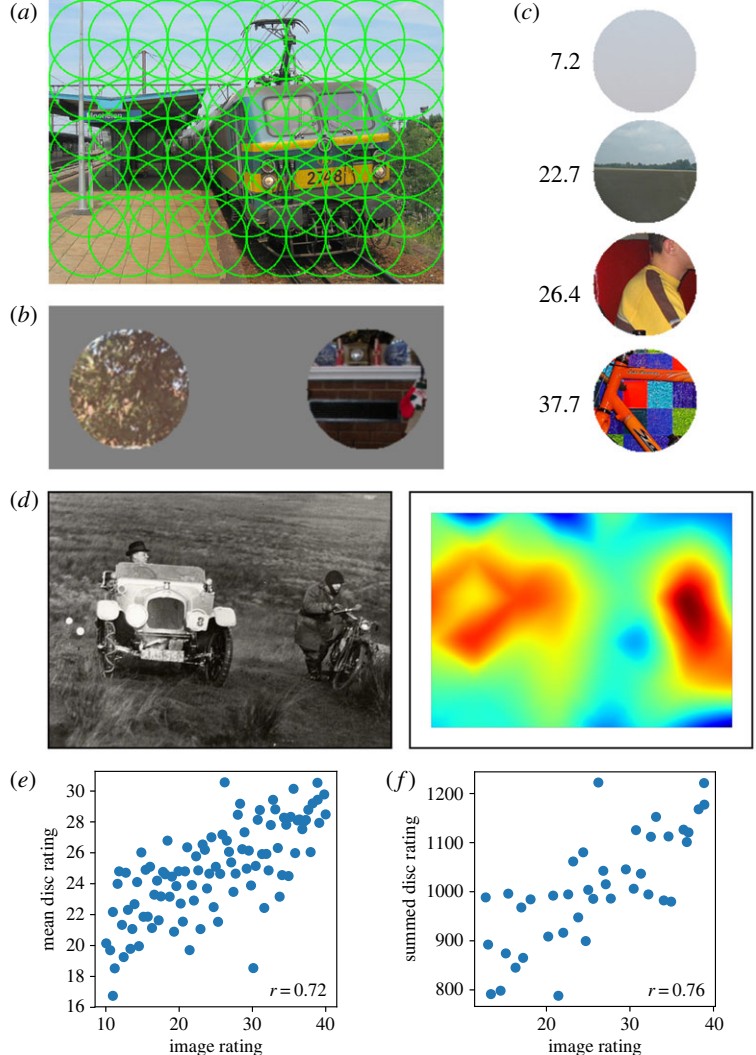

**Figure 5.** Rating complexity of image parts. (*a*) Images were split into overlapping discs. (*b*) Discs were presented to observers in 2AFC comparison trials. (*c*) Discs were rated for perceived complexity using TrueSkill; four example ratings are shown. (*d*) Bicubic interpolation was used to generate whole-image complexity maps directly from disc complexity ratings. (*e,f*) Correlations between whole-image complexity ratings and both averaged and summed ratings of each image's discs. Summed ratings are shown only for the 41 images of size $375 \times 500$.

# 4. Experiment 2: complexity of image parts

In experiment 2 we investigate how the overall percept of an image's complexity relates to locally processed complexity estimates of smaller parts of an image. We split images into small discs of radius 50 pixels, a size which prevents most objects from appearing entirely within one disc. Observers were asked to compare pairs of discs (depicting image fragments) and choose the most complex. We computed complexity scores as in experiment 1. In this way, the ratings of an image's fragments could be compared to that of an entire image. We then evaluated the use of the mean disc complexity and the summed disc complexity as predictors of the entire image's rating.

## 4.1. Material and methods

### 4.1.1. Data generation

From our 4000-image dataset, we sampled 100 images, distributed uniformly according to their complexity rating. Each image was then split into a number of overlapping discs of radius 50 pixels (figure 5 shows an example). The images, which were of different sizes, were broken into between 30 and 50 discs (mean: 38). We obtained 3803 discs in total.

### 4.1.2. Participants

Twenty-three observers (11 male) with a mean age of 28.5 (s.d. 7.9) were recruited from a mailing list run by the UCL Institute of Cognitive Neuroscience. Observers all had normal or corrected-to-normal vision and were asked to have a good night's sleep before attending the experiment.

### 4.1.3. Procedure

Data collection was performed using an online experiment run in a computer laboratory; all participants performed the experiment at the same time. As in our first experiment, disc complexity was judged by 2AFC trials, except that in this case both discs were visible on the screen at the same time. Participants then pressed the Z key to indicate that the first disc was more complex or the M key to select the second disc . We presented 12 practice trials at the beginning of the experiment. Data collection was not divided into blocks, and we allowed free breaks. We collected 51 403 pairwise comparisons of the 3803 discs. Each participant judged on average 2234 comparisons.

### 4.1.4. Post-processing

As in the first experiment, disc comparisons were submitted to TrueSkill, which produced a complexity ranking.

## 4.2. Results

We compared whole-image ratings to each image's mean disc rating and summed disc rating (the summed complexity ratings of all discs making up that image). To prevent image size (which determines disc count) acting as a confound for summed ratings, we calculated summed ratings only for the 41 images of size $375 \times 500$. Results are shown in figure 5. Summed disc ratings were correlated with whole-image complexity ratings with $r = 0.43$. Mean disc ratings were more highly correlated with complexity, with $r = 0.72$. The individual disc ratings were collected by a different set of observers than the whole-image ratings, and the disc observers had never seen the whole images. This suggests that a fair level of prediction of perceived complexity can be obtained from image parts, and that complexity perception thus does not necessarily require whole-image perception or the detection of long-range correlations.

## 4.3. Modelling

Can deep learning models predict individual disc complexity ratings as effectively as they can predict whole-image complexity? To investigate, we used the 3803 rated discs (extracted from 100 images) to train and validate a CNN based on Inception V3 (the model with the best predictive power for the complexity of whole images). We again replaced the final classification layers with five fully-connected layers with {1024, 512, 64, 32, 1} units.

In this case, a validation set was generated by holding back 10% of the 100 images (not of the 3803 discs) for validation. If 10% of discs had been selected, most images would have discs present in both the training and validation sets, allowing the model to exploit similarities in discs taken from the same image. This method ensured that no discs from images featuring in the validation set had been seen during training. The validation set showed a correlation of $r = 0.78$ with perceived complexity ratings.

# 5. Conclusion

We now summarize our results and evaluate the use of predictive models of complexity and of the deep learning approach.

## 5.1. Summary

We present a predictive model demonstrating a high level of correlation ($r = 0.83$) with human-derived complexity ratings on an unseen test set consisting of 400 diverse real-world images. Both our training set and test sets, containing 3600 and 400 images respectively, are larger and more varied than those used to train previous models [26–28].

We show that complexity can be estimated from basic and mid-level features (linear regression gives $r = 0.69$ with human ratings), but can be predicted much more rapidly and effectively using a deep convolutional network ($r = 0.83$). The perceived complexity of 50-pixel-radius discs can also be predicted by a CNN ($r = 0.78$) and mean ground truth disc ratings were a reasonable predictor of whole-image ratings ($r = 0.72$). We used unseen validation sets throughout.

The number and type of objects present in the image influence complexity ratings, and object taggings and counts are among the most informative predictors. PASCAL VOC taggings, via an SVR, predict complexity ratings at $r = 0.49$. These results suggest that object perception plays a role in visual complexity, but does not fully explain it. The higher correlations obtained by our CNN model suggest that it may also be learning texture [54].

Visual complexity is a promising proxy for perceptual load in real-world images and could allow an extension of perceptual load theory [55] to the processing of real-world images. For example, previous research has established a greater likelihood of inattentional blindness and deafness in conditions of high perceptual load [5–7,16,56–58]. Images of higher perceived complexity are likely to place a greater demand on perceptual processing, imposing higher perceptual load.

Our predictive model of complexity could potentially be used to operationalize and quantify perceptual load in the processing of real-world images. Importantly, a method of estimating the level of perceptual load also allows the prediction of the occurrence of inattentional blindness and deafness in tasks involving real-world images [59]. The use of a fast machine learning model also allows complexity prediction in real time; our CNN model can run compactly and efficiently on the central processing unit or graphics processing unit in under 100 ms and, unlike feature-based models, does not require the computation of multiple image statistics using different programs.

In addition, a predictive model of complexity allows investigation of prominent issues in natural scene perception research. For example, the question of whether image complexity impacts gist perception in natural scenes could be approached by varying image complexity and assessing the effect on gist perception.

## 5.2. Utility of a predictive model of complexity

It can be argued that operationalizing visual complexity using a deep network does not bring explanatory value, as one unknown process (complexity judgement in the brain) is replaced by another unknown process (complexity computation in a CNN). We argue that this model has value in showing that a relatively simple feed-forward network can effectively predict image complexity. We provide evidence that complexity can be predicted by a series of stacks of convolutional filters and does not critically depend on recurrent processing or semantic judgements (although these may further improve predictions, and this is an interesting direction for future research).

Importantly, this result demonstrates that it is possible to model the perceived complexity of a diverse set of real-world scenes, achieving a good level of prediction on an unseen validation set. Thus, while our model cannot clarify how the human brain judges complexity, it offers a neural network model which is, if not more biologically plausible, at least more structurally and functionally homogeneous than the sets of feature extractors (using different programming languages and architectures) presented in previous work.

Our model is also useful for generating rapid estimates of the visual complexity of real-world images. Predicting visual complexity has numerous image processing applications in the context of human–machine interfaces HMI. For example, in automated driving, the complexity of visual displays in any non-driving task (such as Internet browsing) could be evaluated in order to estimate the driver's ability to respond to a take-over request signal and reassume control of the vehicle—or instead to experience inattentional blindness or deafness.

## 5.3. Utility of the deep learning approach

We have shown that a CNN is able to estimate complexity directly from pixel values without requiring hand-coded low- or mid-level features. Previous predictive models of visual complexity have drawn upon hard-coded features, assembled into vectors and passed into learning algorithms with small numbers of parameters (regressions, PSO or support vector machines). A CNN, which has access to the value of each pixel in the input, can learn any computable function of pixel-level data providing it is equipped with enough layers and cells. This approach surpasses learning from hard-coded image statistics, where the early features, which cannot be trained, only pass a restricted and fixed subset of pixel-level information to the next level. Our model also has a large number of parameters (16 million) compared to previously

used PSO (12 parameters), regression (hundreds) or SVMs (thousands); it therefore has greater ability to learn complex patterns.

Deep learning also has advantages in the engineering domain; it is fast and energy-efficient, and does not require extensive work to hand-tune input features. We use the same architecture to perform the network's entire prediction process; feature-based approaches require chunks of code, often written in heterogenous languages and having different running times, whereas our model can be homogenously implemented using a deep learning toolkit such as TensorFlow [60] or Torch [61].

## 5.4. Complexity and object recognition

The fact that both our predictive models (whether based on VGG-16 or Inception V3) showed improved performance when their weights were pre-trained on an object recognition task shows that complexity prediction can benefit from transfer learning from object recognition. The PASCAL VOC object taggings are predictive of complexity with $r$ between 0.42 and 0.56, a higher level of correlation than that shown by most image statistics. We also showed that object count, obtained from the Mask R-CNN network, was a better predictor of complexity than any individual image feature. Together, these findings suggest that the nature or number of the objects present in an image contribute to the perception of its complexity.

## 5.5. Future work

There are still unanswered questions concerning the relationship between complexity and clutter. Here we focused on perceived complexity, with our decision task asking the observer which of two images was more complex. Clutter has more usually been defined by direct ratings (which are problematic in terms of bias), rankings (which are less ecologically valid) or proxies such as visual search time. Further work could compare and potentially unify these two ideas.

Ethics. Data collection was ethically approved by University College London (application 9751/002). Informed consent was received from all participants.

Data accessibility. The code and data used to generate the results described in this article is available on Dryad at doi:10.5061/dryad.3fs556j [62] and on Github at https://github.com/fusionlove/image-complexity.

Authors' contributions. F.N. and N.L. conceived and designed the study and wrote the manuscript. F.N. collected the data and carried out modelling and analysis. Both authors gave final approval for publication.

Competing interests. The authors declare that they have no competing interests.

Funding. This research was supported by Jaguar Land Rover and the EPSRC grant no. EP/N012089/1 as part of the jointly funded Towards Autonomy: Smart and Connected Control (TASCC) programme.

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
