## [Reviewer comments · Royal Society Open Science]

Review History

RSOS-191487.R0 (Original submission)

Review form: Reviewer 1

Is the manuscript scientifically sound in its present form?

No

Are the interpretations and conclusions justified by the results?

No

Is the language acceptable?

Yes

Do you have any ethical concerns with this paper?

No

Have you any concerns about statistical analyses in this paper?

No

Recommendation?

Major revision is needed (please make suggestions in comments)

Comments to the Author(s)

Please find my review comments in the attached review document (Appendix A).

Review form: Reviewer 2

Is the manuscript scientifically sound in its present form?

No

Are the interpretations and conclusions justified by the results?

No

Is the language acceptable?

Yes

Do you have any ethical concerns with this paper?

No

Have you any concerns about statistical analyses in this paper?

Yes

Recommendation?

Major revision is needed (please make suggestions in comments)

Comments to the Author(s)

In this paper, the authors operationalize perceptual load for natural images in terms of visual complexity and build a deep neural network feature-based model to predict human-derived scores of visual complexity. Although the authors have collected a large dataset of human behavioral measurements of image complexity and tested various feature-based representations to predict the same, I have a few major concerns about this study which I am noting down below:

1. The link between perceptual load and complexity is not clear. Previous literature suggests that perceptual load is a function of object/feature density/clutter in the context of visual search (for example, visual search is harder for heterogenous distracters). However, the fact that visual search times and the behavioral complexity scores are not strongly correlated indicates that the authors could be measuring something else altogether.
2. It looks like each image pair was rated only once. If so, it raises the possibility that some of the measurements could be erroneous.
3. What is the reliability of the data? Is there a measure of explainable variance for the dataset? Because, it becomes very difficult to understand where each of the models stand in terms of 'predictive power' without an actual measure of split-half reliability.
4. What is the rationale behind using a comparative rating task on pairs of images as opposed to using a likert-like rating task on individual images? It seems like the authors measured relative complexity only to then convert them to image-wise ratings using the TrueSkill algorithm. This choice needs to be justified and elaborated further.
5. "The 2AFC judgements matched the TrueSkill ratings in 76.1% of comparisons" -- How does this number (76.1%) translate to actual complexity values? Are there images for which complexity values are not reliable? Were such images discarded from further analysis?
6. For the Vgg-16 architecture, why use 6 fully-connected layers instead of going directly from 1024-layer to a scalar output? And, why is the range of the scalar output limited to [0,1]? How is the TrueSkill-derived complexity scores squished to a range [0,1] when they are distributed with a mean of 25?
7. In case of the linear regression model on raw pixel vectors, it is not clear what the linear regression model is. Aren't the number of pixels (variables) way more than the number of

equations (= # of images)? Did the authors use some constraints on the weights (lasso or ridge regression)? Also, is the validation set same for all models? Right now, the manuscript reads like a list of things the authors tried out without elaborating on the how's and why's of either the analyses or results.

8. Why did the authors choose windows of 50px radius for the part-based analysis? I think that an eccentricity-based sampling approach could make more sense in this scenario.

9. There is previous literature on modeling visual clutter which is relevant to this work. Here are a few examples --

a. Yu, C. P., Samaras, D., & Zelinsky, G. J. (2014). Modeling visual clutter perception using proto-object segmentation. *Journal of vision*, 14(7), 4-4.

b. Deza, A., & Eckstein, M. (2016). Can peripheral representations improve clutter metrics on complex scenes?. In *Advances in Neural Information Processing Systems* (pp. 2847-2855).

Decision letter (RSOS-191487.R0)

12-Dec-2019

Dear Dr Nagle,

The editors assigned to your paper ("Predicting human complexity perception of real-world scenes") have now received comments from reviewers. We would like you to revise your paper in accordance with the referee and Associate Editor suggestions which can be found below (not including confidential reports to the Editor). Please note this decision does not guarantee eventual acceptance.

Please submit a copy of your revised paper before 04-Jan-2020. Please note that the revision deadline will expire at 00.00am on this date. If we do not hear from you within this time then it will be assumed that the paper has been withdrawn. In exceptional circumstances, extensions may be possible if agreed with the Editorial Office in advance. We do not allow multiple rounds of revision so we urge you to make every effort to fully address all of the comments at this stage. If deemed necessary by the Editors, your manuscript will be sent back to one or more of the original reviewers for assessment. If the original reviewers are not available, we may invite new reviewers.

If your study uses humans or animals please include details of the ethical approval received, including the name of the committee that granted approval. For human studies please also detail

whether informed consent was obtained. For field studies on animals please include details of all permissions, licences and/or approvals granted to carry out the fieldwork.

- Data accessibility

If you wish to submit your supporting data or code to Dryad (<http://datadryad.org/>), or modify your current submission to dryad, please use the following link:
<http://datadryad.org/submit?journalID=RSOS&manu=RSOS-191487>

- Competing interests

- Authors' contributions

- Acknowledgements

- Funding statement

Kind regards,

Anita Kristiansen

Editorial Coordinator

on behalf of Dr Narayanan Srinivasan (Associate Editor) and Essi Viding (Subject Editor)
 openscience@royalsociety.org

Associate Editor's comments (Dr Narayanan Srinivasan):

Two experts have now commented on the paper. They have expressed some concerns and have raised questions. The authors are requested to address all the comments and questions point by point.

Reviewers' Comments to Author:

Reviewer: 1

Comments to the Author(s)

Please find my review comments in the attached review document.

Reviewer: 2

Comments to the Author(s)

In this paper, the authors operationalize perceptual load for natural images in terms of visual complexity and build a deep neural network feature-based model to predict human-derived scores of visual complexity. Although the authors have collected a large dataset of human behavioral measurements of image complexity and tested various feature-based representations to predict the same, I have a few major concerns about this study which I am noting down below:

1. The link between perceptual load and complexity is not clear. Previous literature suggests that perceptual load is a function of object/feature density/clutter in the context of visual search (for example, visual search is harder for heterogenous distracters). However, the fact that visual search times and the behavioral complexity scores are not strongly correlated indicates that the authors could be measuring something else altogether.
2. It looks like each image pair was rated only once. If so, it raises the possibility that some of the measurements could be erroneous.
3. What is the reliability of the data? Is there a measure of explainable variance for the dataset? Because, it becomes very difficult to understand where each of the models stand in terms of 'predictive power' without an actual measure of split-half reliability.
4. What is the rationale behind using a comparative rating task on pairs of images as opposed to using a likert-like rating task on individual images? It seems like the authors measured relative complexity only to then convert them to image-wise ratings using the TrueSkill algorithm. This choice needs to be justified and elaborated further.
5. "The 2AFC judgements matched the TrueSkill ratings in 76.1% of comparisons" -- How does this number (76.1%) translate to actual complexity values? Are there images for which complexity values are not reliable? Were such images discarded from further analysis?
6. For the Vgg-16 architecture, why use 6 fully-connected layers instead of going directly from 1024-layer to a scalar output? And, why is the range of the scalar output limited to [0,1]? How is the TrueSkill-derived complexity scores squished to a range [0,1] when they are distributed with a mean of 25?
7. In case of the linear regression model on raw pixel vectors, it is not clear what the linear regression model is. Aren't the number of pixels (variables) way more than the number of equations (= # of images)? Did the authors use some constraints on the weights (lasso or ridge regression)? Also, is the validation set same for all models? Right now, the manuscript reads like a list of things the authors tried out without elaborating on the how's and why's of either the analyses or results.
8. Why did the authors choose windows of 50px radius for the part-based analysis? I think that an eccentricity-based sampling approach could make more sense in this scenario.
9. There is previous literature on modeling visual clutter which is relevant to this work. Here are a few examples --

- a. Yu, C. P., Samaras, D., & Zelinsky, G. J. (2014). Modeling visual clutter perception using proto-object segmentation. *Journal of vision*, 14(7), 4-4.
- b. Deza, A., & Eckstein, M. (2016). Can peripheral representations improve clutter metrics on complex scenes?. In *Advances in Neural Information Processing Systems* (pp. 2847-2855).

Author's Response to Decision Letter for (RSOS-191487.R0)

See Appendix B.

RSOS-191487.R1 (Revision)

Review form: Reviewer 1

Is the manuscript scientifically sound in its present form?

Yes

Are the interpretations and conclusions justified by the results?

Yes

Is the language acceptable?

Yes

Do you have any ethical concerns with this paper?

No

Have you any concerns about statistical analyses in this paper?

No

Recommendation?

Accept with minor revision (please list in comments)

Comments to the Author(s)

It would have been nicer if the authors had included actual SIFT/SURF features instead of interest point counts. Surely the number of interest points alone will not give enough information about local patches as the actual features that are commonly used once these interest points are detected.

Decision letter (RSOS-191487.R1)

10-Mar-2020

Dear Dr Nagle:

On behalf of the Editors, I am pleased to inform you that your Manuscript RSOS-191487.R1 entitled "Predicting human complexity perception of real-world scenes" has been accepted for publication in *Royal Society Open Science* subject to minor revision in accordance with the referee suggestions. Please find the referees' comments at the end of this email.

The reviewers and Subject Editor have recommended publication, but also suggest some minor revisions to your manuscript. Therefore, I invite you to respond to the comments and revise your manuscript.

- Ethics statement

- Data accessibility

If you wish to submit your supporting data or code to Dryad (<http://datadryad.org/>), or modify your current submission to dryad, please use the following link:
<http://datadryad.org/submit?journalID=RSOS&manu=RSOS-191487.R1>

- Competing interests

- Authors' contributions

- Acknowledgements

- Funding statement

Please note that we cannot publish your manuscript without these end statements included. We have included a screenshot example of the end statements for reference. If you feel that a given

heading is not relevant to your paper, please nevertheless include the heading and explicitly state that it is not relevant to your work.

Because the schedule for publication is very tight, it is a condition of publication that you submit the revised version of your manuscript before 19-Mar-2020. Please note that the revision deadline will expire at 00.00am on this date. If you do not think you will be able to meet this date please let me know immediately.

on behalf of Dr Narayanan Srinivasan (Associate Editor) and Essi Viding (Subject Editor)
openscience@royalsociety.org

Associate Editor Comments to Author (Dr Narayanan Srinivasan):

The reviewer is generally satisfied with the revision but has one more comment. The authors are requested to address that comment and submit the final version before final acceptance.

Reviewer comments to Author:

Reviewer: 1

Comments to the Author(s)

It would have been nicer if the authors had included actual SIFT/SURF features instead of interest point counts. Surely the number of interest points alone will not give enough information about local patches as the actual features that are commonly used once these interest points are detected.

Author's Response to Decision Letter for (RSOS-191487.R1)

See Appendix C.

Decision letter (RSOS-191487.R2)

07-Apr-2020

Dear Dr Nagle,

It is a pleasure to accept your manuscript entitled "Predicting human complexity perception of real-world scenes" in its current form for publication in Royal Society Open Science.

on behalf of Dr Narayanan Srinivasan (Associate Editor)
openscience@royalsociety.org

Follow Royal Society Publishing on Twitter: [@RSocPublishing](https://twitter.com/RSocPublishing)
Follow Royal Society Publishing on Facebook:
<https://www.facebook.com/RoyalSocietyPublishing.FanPage/>
Read Royal Society Publishing's blog: <https://blogs.royalsociety.org/publishing/>

Appendix A

Predicting human complexity perception of real-world scenes

Fintan Nagle , Nilli Lavie

Summary: Authors attempt to develop a image-computable measure of visual complexity, as a proxy for perceptual load in visual tasks. They do this by collecting 75,020 complexity ratings on image pair comparisons using 4000 images drawn from the Pascal VOC image set. Since the authors don't sample the complete 4000 C 2 pairs for each participant, they use the Trueskill algorithm that estimates the all-pairwise distance distribution from a limited set of distance observations. They report that this method yields an agreement of 76.1% with observed pair-wise rankings. Subsequently, they employ these predicted scores to train a deep convolutional network in an end-to-end fashion and show that the trained networks out-performs many existing baseline feature schemes when it comes to predicting perceived visual complexity. Authors also perform additional experiments that suggest that visual complexity in an image is largely governed by the visual statistics of local regions/patches and this is again done by obtaining 2AFC visual complexity ratings, in a similar fashion as for the whole image. Overall, the paper is written clearly and in a linear fashion and is easy to read. That said, it would have been nice to have some more information about whether cross-validation had been employed in the regression model. Overall I found the paper interesting and timely although I have concerns that I am outlining below.

My concern here is that operationalizing visual complexity using CNNs and not doing proper controls, ends up replacing one unknown (computation of visual complexity in human brain), with another (computation of the same in CNN). Some of this seems to be happening in this paper as well.

Proto-objects (<https://www.sciencedirect.com/science/article/pii/S004269891300240X>), object parts and clutter (<https://link.springer.com/article/10.3758/s13414-017-1359-9>) have all been shown to play a role in perceptual load. The author's own second experiment also indicates the same. It seems very odd then that none of the good intermediate level features (SIFT, MSER, SURF... all available in easily implemented libraries eg; <http://www.vfeat.org/>) have been modelled in Table 1. Is it possible that a combination of such intermediate level features will be equally good or better than the CNN implementation? Besides, these mid-level feature operators are defined independent of a specific image set and should give greater generalisation. It'll be great if these can be included in Table 1 as well as in the combined model that uses all available parameters. I would definitely ask the authors to include these mid-level operators, else their results in Table 1 seem incomplete.

Does the regression model work well because of overfitting? Table 2. What are the weights of such a regression model? That can give a hint about what actually goes into defining visual complexity.

How exactly are the 75,020 pairs distributed over 62 pairs?

How exactly was the rating modelled in Trueskill? No details are given. Is each person pitted against others?

A general question is how appropriate is Trueskill in modeling pair-wise complexity distribution. The algorithm was originally developed to compare video game players for their skill. So how does this translate to a good algorithm for this purpose. It would have been nice if a more substantial argument had been given for this choice. Or some meaningful comparison perhaps. Is it possible that a better choice, would have then led to the CNN doing better as well? For example, how does the classic ELO rating system do in this context? Perhaps the authors could give that performance against Trueskill's 76% accuracy. How about the Glicko rating system that is used to model 1-vs-1 player games? Here again it is not clear how exactly the authors modelled their annotators in the Trueskill system.

I hope that the authors intend to publish their trained CNN models and behavioural data in the public domain on eventual acceptance of this manuscript. I couldn't get access to the dryad DOI mentioned in the submission.

Dear Editor,

We thank the reviewers for their careful reading of and constructive feedback on our manuscript. We have addressed all the points raised and have performed all the requested additional analyses, as we detail below.

Reviewer 1

Since the authors don't sample the complete 4000 C 2 pairs for each participant, they use the Trueskill algorithm that estimates the all-pairwise distance distribution from a limited set of distance observations

We did not aim to characterise all the pairs, but instead to characterise the perceived complexity of each individual image. TrueSkill does this by taking into account the results of all the comparisons. We validate this method by showing that it explains 76.1% of comparisons (no model could explain 100% of comparisons from a group of different observers, due to noise and inter-observer disagreement).

We have made this clarification in the introduction:

Here we aim to predict perceived image complexity in order to use it as a proxy for perceptual load in natural scene perception. We collected 75,020 pairwise 2AFC complexity judgements ("which image is more complex?") over a set of 4000 natural scene images from the PASCAL VOC dataset \cite{everingham2010pascal}, widely used in computer vision research on object recognition. From these comparisons, we generated a complexity ranking over the images, assigning a complexity score to each one. We then applied a deep learning approach, training a convolutional neural network to predict the complexity scores of each of these varied real-world scenes.

We have also made this clarification in the description of Experiment 1:

Here we trained a deep convolutional network to predict the perceptual complexity of 4,000 images of a variety of natural scenes, aiming to make accurate predictions while avoiding overfitting. We aimed to predict the complexity of each image, rather than the outcomes of individual 2AFC comparisons or the full 4000-by-4000 distance matrix.

it would have been nice to have some more information about whether cross-validation had been employed in the regression model.

All models were validated on a randomly chosen validation set. Cross-validation could not be employed when training CNNs due to the time taken to re-train these models. However, we used a randomly chosen unseen validation set of 400 images (10% of the data), so we are confident that it is both representative of the data and did not pass any information to the network during training which could allow it to overfit to the validation set.

We have added additional analyses (described later) to cross-validate and regularise the regression model.

My concern here is that operationalizing visual complexity using CNNs and not doing proper controls, ends up replacing one unknown (computation of visual complexity in human brain), with another (computation of the same in CNN). Some of this seems to be happening in this paper as well.

We have shown that the concept of visual complexity we are measuring is meaningful, coherent among observers, and useful for predicting perceptual judgements. It is true that there is no high-level conceptual model of what computations are taking place in the CNN, but this does not detract from its usefulness as a model.

We ensured that the model was not overfit by using an unseen validation set of 400 images.

We have clarified this in the discussion:

Utility of a predictive model of complexity

It can be argued that operationalising visual complexity using a deep network does not bring explanatory value, since a complex unknown process (complexity judgements in the brain) is replaced by another unknown process (complexity computation in a CNN). We argue that this model has value in showing that a relatively simple feedforward network can effectively predict image complexity. We provide evidence that complexity can be predicted by a series of stacks of convolutional filters and is not dependent on, for example, recurrent processing or semantic judgements (although these may improve predictions further, and this is an interesting direction for future research).

Importantly, this result demonstrates that it is possible to model perceived complexity of a diverse set of real-world scenes, achieving a good level of prediction on an unseen validation set while avoiding overfitting. Thus, while our model cannot clarify how the human brain computes complexity, it offers a neural network model which is, if not more biologically plausible, at least more structurally and functionally homogenous than the sets of feature extractors (using different programming languages and architectures) presented by previous work.

Our model is also useful for generating rapid estimates of the visual complexity of real-world images. Predicting visual complexity has numerous image

processing applications in the context of human-machine interfaces (HMI). For example, in automated driving, the complexity of visual displays in any non-driving task (such as internet browsing) could be evaluated in order to estimate the driver's ability to respond to a take-over request signal (or instead experience inattentional blindness or deafness) and reassume control of the vehicle.

The author's own second experiment also indicates the same. It seems very odd then that none of the good intermediate level features (SIFT, MSER, SURF... all available in easily implemented libraries eg; <http://www.vlfeat.org/>) have been modelled in Table 1. Is it possible that a combination of such intermediate level features will be equally good or better than the CNN implementation? Besides, these mid-level feature operators are defined independent of a specific image set and should give greater generalisation. It'll be great if these can be included in Table1 as well as in the combined model that uses all available parameters. I would definitely ask the authors to include these mid-level operators, else their results in Table 1 seem incomplete.

We thank the reviewer for raising this point and have now computed SIFT, SURF and MSER features for each image, and introduced their counts into our scalar feature model (which, as a regression, could only accept one data point per feature per image). To attempt to characterise the nature of the features as well as their count, we submitted the SIFT and SURF features to classifiers. This could not be done for MSER regions, which are simply mappings from pixels to region ID. The new analyses are reported in Section 3c.

Does the regression model work well because of overfitting? Table 2. What are the weights of such a regression model? That can give a hint about what actually goes into defining visual complexity.

This model has been trained a large number of times with cross-validation. At each step, accuracy was tested on an unseen validation set, this makes it highly unlikely that our model works well because of overfitting. We also added ridge and lasso regularisation (reported in Section 3c), and these methods did not cause accuracy drops, indicating that the original model was not overfit.

We have added the weights of the trained regression model to Table 1, and we thank the reviewer for bringing this point up.

How exactly are the 75,020 pairs distributed over 62 pairs?

We assume the reviewer means "62 observers" and have added this clarification to the methods to clarify this:

Pairwise comparisons were randomly distributed across participants, so that image presentation counts were approximately equal per each participant.. (The counts were approximate, rather than perfectly balanced due to the crowdsourced nature of the experimental platform.)

How exactly was the rating modelled in Trueskill? No details are given. Is each person pitted against others?

We have added a clarification on TrueSkill (Section 2). We also added the following clarification to the methods under post-processing:

Post-processing. The web interface provided us with a list of comparisons, each one effectively a competition in complexity between two images. Each image participated in approximately 37 comparisons. These comparisons were used as input to the TrueSkill algorithm, which uses a Bayesian framework to assign a score distribution to each image based on its performance in each competition. We used the mean of this distribution as that image's complexity rating. Individual observer results were not modelled and no competitions between observers were set up.

A general question is how appropriate is Trueskill in modeling pair-wise complexity distribution. The algorithm was originally developed to compare video game players for their skill. So how does this translate to a good algorithm for this purpose. It would have been nice if a more substantial argument had been given for this choice. Or some meaningful comparison perhaps. Is it possible that a better choice, would have then led to the CNN doing better as well? For example, how does the classic ELO rating system do in this context ? Perhaps the authors could give that performance against Trueskill's 76% accuracy. How about the Glicko rating system that is used to model 1-vs-1 player games? Here again it is not clear how exactly the authors modelled their annotators in the Trueskill system.

As mentioned in our earlier response, we have now added a detailed description of how complexity comparisons were passed to TrueSkill.

Our choice of TrueSkill was informed by the following factors. Firstly, it is a robust, well-tested, state-of-the-art model. Secondly, it does much more computation than simpler models (such as Thurstone's model or win counting). Thirdly, it is a development and a generalisation of the ELO rating system and is widely accepted to be superior on large datasets. We have noted these three factors as follows:

There are many approaches to generating a ranking from paired comparisons, going back to Thurstone \cite{thurstone1927law}. Game theoretic techniques, which model comparisons as competitions between images, are often used to assign a score to each competitor; examples are the Elo rating system \cite{glickman1999rating} and its extension the Glicko system \cite{glickman1995glicko}. We used TrueSkill, a state-of-the-art method which also generalises the Elo method and is newer than the Glicko method.

I hope that the authors intend to publish their trained CNN models and behavioural data in the public domain on eventual acceptance of this manuscript. I couldn't get access to the dryad DOI mentioned in the submission.

We are not sure why there was no access; perhaps this is because the paper is still under review. We will make sure this Dryad repository is activated. You can also access the trained models and data at <https://github.com/fusionlove/image-complexity>; we have added this link to the paper.

Reviewer 2

the authors operationalize perceptual load for natural images in terms of visual complexity and build a deep neural network feature-based model

While we agree that we “**operationalize perceptual load for natural images in terms of visual complexity**” with respect to the comment that we “**build a deep neural network feature-based model**” we clarify in the paper that we make a distinction between feature-based models (which rely on a sparse set of features, possibly hand-coded) and pixel-based deep learning models (which can learn a large number of arbitrary stacked convolutional filters). We refer to these as stacked filters rather than features to underline that they are multilevel and include other computations such as max-pooling, unlike the simpler features (such as lines, edges, templates and SIFT) used in computer vision.

We argue that deep learning models are better than feature-based models because they have the opportunity to take every single pixel of the input into account, there are no restrictions on the filters they can learn, and they can learn a large number of filters and then combine the results to compute a final output value.

The link between perceptual load and complexity is not clear. Previous literature suggests that perceptual load is a function of object/feature density/clutter in the context of visual search (for example, visual search is harder for heterogenous distracters). However, the fact that visual search times and the behavioral complexity scores are not strongly correlated indicates that the authors could be measuring something else altogether.

We have revised the paper discussion to make the link clearer, but overall, we consider this paper as the first to start relating the two and we refer in the summary to additional behavioural work that further establishes the link between complexity and perceptual load. Nevertheless, as the first step in the present study, we were cautious to refer to complexity as a possible proxy for perceptual load, rather than to equate the two.

As for the relationship to visual search RT we have expanded our discussion of the results in the revised section g as follows:

(g) Complexity and visual search reaction time

We compared complexity ratings to human visual search reaction times obtained from Ionescu et al. [50]; there was a fairly low correlation between the two scores ($r = .37$). This not surprising since the two tasks (judgements of complexity and search for the presence of a particular object) were different to

each other. Indeed correlations of this size are typically found among different tasks which share a common cognitive factor, specifically one that relates to the perceptual processing demands of the task [52,53].

Moreover, In busy real-world scenes, visual search RT is expected to depend not only on image complexity, but also on other factors that are specific to the relations between the search target and the scene characteristics - salience, for example. It is easy to imagine a highly complex image with a very short visual search time (searching for a bright red balloon in a busy grey cityscape) - as well as an image of low complexity with a long visual search time (searching for a snowy owl against a snowy background). Semantic contextual factors can play a role too: for example, RT to detect that a sofa is absent in an image of an outdoor scene should be faster than in an image of an indoors scene irrespective of their level of complexity).

In addition, in section (a) of the Conclusions we refer to previous literature that has related perceptual load to various tasks (not just visual search) and across various perceptual complexity manipulations, not just set size or heterogeneity. An example is the requirement to process feature conjunctions [high load] versus single features [low load] from the very same items). These manipulations have served to clarify that the concept of perceptual load is more general than that of set size or heterogeneity.

it looks like each image pair was rated only once. If so, it raises the possibility that some of the measurements could be erroneous.

We did not aim to characterise each image pair; this would be intractable, since there are $4000 \times 4000 = 16$ million image pairs. It would not be directly informative about the complexity of each image either. Since our was to characterise each image this meant that we needed to calculate a stable, confident complexity estimate for each image. We did this by performing 75,020 comparisons, meaning that each image participated in 37 comparisons on average. We validated this by checking for convergence in TrueSkill ratings (continuing data acquisition until ratings converged) and by checking that ratings explained 76.1% of human comparisons.

It is impossible to explain 100% of a group's comparisons, since some are random and observers' criteria differ. We are not aiming to predict individual decisions but collective complexity perception of images.

What is the reliability of the data? Is there a measure of explainable variance for the dataset? Because, it becomes very difficult to understand where each of the models stand in terms of 'predictive power' without an actual measure of split-half reliability.

We believe that image complexity ratings are highly reliable firstly since they explain 76.1% of human comparisons, and secondly since we collected data until the ratings converged (in other words, additional ratings would not have changed the correlation value).

We also believe that the model prediction accuracies are also highly reliable, since all accuracies we report were obtained from a validation set unseen during training. All regressions have been cross-validated. Model ratings are also highly generalisable, since we used a large set of 4000 varied, real-world images.

It was not clear to us how to apply split-half reliability in this case.

What is the rationale behind using a comparative rating task on pairs of images as opposed to using a likert-like rating task on individual images? It seems like the authors measured relative complexity only to then convert them to image-wise ratings using the TrueSkill algorithm. This choice needs to be justified and elaborated further.

We agree and have added the following clarifications to the manuscript:

(Section 1 C) - Our approach

For data collection, most of the studies just described used Likert scales. Here we used 2-alternative forced choice (2AFC) paired comparisons; observers were presented with pairs of images and asked to indicate which was more complex. The 2AFC approach has a few advantages over Likert-like ratings. Firstly, it is less subject to response bias. For example, a conservative observer may rank all images lower than a non-conservative observer, but this would not affect their 2AFC choice, which is based on judgement of relative complexity within a pair. Secondly, the method of 2AFC paired comparisons is more resistant to changes in criterion over the course of exposure to more images in the experiment since observers are forced to choose among images of each pair rather than relating each image to the increasing number of previously rated images.

We therefore generated complexity ratings for 4,000 images by presenting pairs of images and requesting observers to make 2AFC judgement comparing the images in the pair to each other. This allowed us to collect 75,020 comparisons which we then converted into a complexity score for each image.

(Section 2 A)

The web interface provided us with a list of comparisons, each one effectively a competition in complexity between two images. These comparisons were used as input to the TrueSkill algorithm, which uses a Bayesian framework to assign a score distribution to each image based on its performance in each competition. We used the mean of this distribution as that image's complexity rating. Individual observer results were not modelled and no competitions between observers were set up.

Section 2: Introduction

There are many approaches to generating a ranking from paired comparisons, going back to Thurstone \cite{thurstone1927law}. Simple methods count the number of paired comparisons won \cite{shah2017simple}. We used TrueSkill, a state-of-the-art method which generalises the Elo system by modelling comparisons as competitions between images.

The 2AFC judgements matched the TrueSkill ratings in 76.1% of comparisons” -- How does this number (76.1%) translate to actual complexity values? Are there images for which complexity values are not reliable? Were such images discarded from further analysis?

We considered a pair of image scores to explain a comparison if the most complex image had the higher score. This was true in 76.1% of cases.

Since all images participated in 30+ comparisons, all complexity values are reliable; this is indicated by the sigma parameter given by TrueSkill being small. {I was not sure what you meant by complexity distribution? Therefore no image was discarded from further analysis.

For the Vgg-16 architecture, why use 6 fully-connected layers instead of going directly from 1024-layer to a scalar output? And, why is the range of the scalar output limited to [0,1]? How is the TrueSkill-derived complexity scores squished to a range [0,1] when they are distributed with a mean of 25?

Inputs and outputs were not scaled - inputs were simply shifted to bring their means closer to zero. We have clarified this as follows in Section 3b.

For the VGG-16 architecture, we removed the final softmax (discrete classification) layer, replacing it with six fully-connected layers with {512, 256, 128, 64, 32, 1} neurons and terminating in a ReLU unit outputting a scalar complexity estimate. Neither inputs nor outputs were scaled; as is standard practice, mean input pixel values were shifted closer to zero by subtracting the means of the ILSVRC colour channels (103.939, 116.779 and 123.680 for R, B and G respectively). For the Inception V3 network [32], we removed the final classification layers and replaced them with with five fully-connected layers with {1024, 512, 64, 32, 1} neurons, the final layer outputting a scalar complexity estimate.

Why use 6 fully-connected layers instead of going directly from 1024-layer to a scalar output?

Firstly as we mention in the added clarifications, the final softmax layer provides discrete classification and so we replaced it with six fully-connected layers with {512, 256, 128, 64, 32, 1} neurons and terminating in a ReLU unit outputting a scalar complexity estimate. Secondly, it is common practice to use a series of fully-connected layers with gradually decreasing unit counts. This technique is standard in deep learning, since it allows the network to perform computations not possible in a single layer. If the network led directly from the 1024-unit layer to the output, the output would simply be a weighted sum of these 1024-unit layers; this has far less computational power than a series of layers.

In case of the linear regression model on raw pixel vectors, it is not clear what the linear regression model is. Aren't the number of pixels (variables) way more than the number of equations (= # of images)? Did the authors use some constraints on the weights (lasso or ridge regression)? Also, is the validation set same for all models?

The number of variables is indeed much higher than the number of inputs. Overfitting is prevented by the use of an unseen validation set; since the exemplars used for validation were not seen during training, high accuracy on the validation set cannot be caused by overfitting. A randomly chosen validation set was used for each model, to reduce the probability in each case of having chosen a very easy validation set.

We have added additional analyses describing lasso and ridge regression, which are described as follows:

(d) Predicting complexity from scalar image features

We tested the usefulness of the feature combination approach by submitting these 38 features to three learners: linear regression, a support vector machine and a 3-layer feedforward neural network.

Straightforward linear regressions, evaluated on a randomly selected unseen 10% validation set, achieved on average $r = 0.65$. Since the validation set was always unseen during training, this model's accuracy has not been inflated by overfitting. We also conducted a full leave-one-out cross-validation, obtaining $r = 0.70$ over 4000 iterations. The observed accuracy gain of 5 percentage points shows that a larger training set led to more accurate predictions.

To further rule out overfitting effects, we trained models with lasso (L1) and ridge (L2) regularisation. In each case, we trained the model on 90% of the data, performing a grid search across 10 values of α from 10⁻¹⁵ to 20, converging on $\alpha = 0.0001$ (lasso regression) and $\alpha = 0.01$ (ridge regression). This process was repeated 10 times. Average accuracy on the held-back validation sets was $r = 0.693$ for lasso regression and $r = 0.694$ for ridge regression.

These results show convincingly that linear regression from scalar image statistics can predict real-world scene complexity at approximately $r = 0.69$. All accuracies were evaluated on unseen training data and a full leave-one-out cross-validation showed the highest correlation with human complexity ratings ($r = 0.70$).

Results are shown in Table 2. As shown, linear regression achieved the highest results, at $r = 0.65$. Predictions were thus improved by pooling multiple features, but were still of low accuracy. The regression coefficients for each image feature, averaged over the reported 4000 runs of leave-one-out cross-validation, were reported in Table 1.

(e) Predicting complexity by characterising mid-level features

Mid-level features such as SIFT and SURF aim to characterise the local neighbourhoods of keypoints in an image. We have shown that counting

extracted keypoints gives some information about an image's complexity. Are the values of these descriptors also useful?

To investigate, we generated SIFT and SURF descriptions of constant length by sampling 500 keypoints from each image, with replacement. This allowed us to analyse the character of the keypoints while controlling for their number. We assembled each image's keypoints into a vector of length $500 \times 128 = 64,000$ (SIFT) or $500 \times 64 = 32,000$ (SURF). Image vectors were then used to train a support vector regression on a randomly selected 90% training set.

Why did the authors choose windows of 50px radius for the part-based analysis? I think that an eccentricity-based sampling approach could make more sense in this scenario.

An eccentricity-based sampling approach, while interesting, rests on its own assumptions and this would have detracted from the main novel focus of the present work. We therefore chose one disc size which was small enough to break up most objects and have added the following clarification:

We added the following clarification:

In Experiment 2 we investigate how the overall percept of an image's complexity relates to locally processed complexity estimates of smaller parts of an image. We split images into small discs of radius 50 pixels, a size which prevents most objects from appearing entirely within one disc. Observers were asked to compare pairs of discs (depicting image fragments) for complexity. We computed complexity scores as in Experiment 1. In this way, the ratings of an image's fragments could be compared to that of an entire image. We then evaluated the utility of the mean disc complexity and the summed disc complexity as predictors of the entire image's rating.

There is previous literature on modeling visual clutter which is relevant to this work. Here are a few examples --

- a. Yu, C. P., Samaras, D., & Zelinsky, G. J. (2014). Modeling visual clutter perception using proto-object segmentation. *Journal of vision*, 14(7), 4-4.
- b. Deza, A., & Eckstein, M. (2016). Can peripheral representations improve clutter metrics on complex scenes?. In *Advances in Neural Information Processing Systems* (pp. 2847-2855).

We thank the reviewer for bringing these interesting papers to our attention and have added the following section on complexity and clutter:

(a) Complexity and visual clutter

The concept of visual complexity is related to that of visual clutter, which Rosenholtz defines as "the state in which excess items, or their representation or organisation, lead to a degradation of performance at some task." A key model of visual clutter is the Rosenholtz feature congestion model [24], which computes local image statistics such as colour variance. It is based on hand-crafted features rather than learned parameters.

This model has been frequently extended. Deza and Eckstein [?], for example, presented an adaptation of the Rosenholtz feature congestion model to a foveated context, using increased resolution at the image centre. This model was validated against visual search reaction time judgements.

Yu et al. [?] proposed a proto-object model of visual clutter based on superpixels. Here, images were segmented into groups of similar pixels using methods such as SLIC [?], then clutter was predicted from region count. Observers also ranked a set of 90 images in order of perceived clutter. The model predicted the ranks quite well, achieving a Spearman's rho of 0.81 .

Is clutter the same concept as complexity? Is clutter the same concept as complexity? While the concept of 'excess items' appears related to that of complexity, it is less clear that the disorderly organisation that reflects higher clutter always indicates higher complexity. One can imagine a scene of high complexity which, due to an ordered arrangement of objects, does not appear cluttered.

Visual clutter has been studied in close relationship with the psychophysics of visual search, with examples of degradation in object recognition or visual search, such as crowding. Measures of visual clutter are therefore informed primarily by performance drops in visual search, usually with simple, well-segmented targets and distractors although Yu et al have presented an important extension to proto-objects.

In contrast, our image comparison task attempted to measure the perceived visual complexity of diverse real-world images. Our complexity ratings were not obtained by accuracy or RT measurements of recognition or search. Neither did they correlate well with visual search reaction times on the same images (see modelling results). Our complexity ratings, therefore, are informed primarily by observers' idea of visual complexity as a descriptor.

We also note that "clutter" is more generally used in natural language to denote objects (e.g. "a cluttered room") and is not as applicable to natural scenes. "Complex" and "simple", however, apply naturally to any visual stimulus without connoting a profusion of objects.

In addition we have correlated our complexity ratings with statistics from the Rosenholtz feature congestion model as well as with object counts and region counts obtained by machine learning and image processing.

We would like to thank the reviewers once again for their detailed feedback. We believe that the revised paper has improved as a result of addressing all the points raised (including the added analyses) and hope that you will now find it suitable for publication in Royal Society: Open Science.

Sincerely,

Fintan Nagle and Nilli Lavie

Dear Editor,

We thank the reviewers and yourself for the positive evaluation of our revision. Please find below our response to the final comment raised.

Reviewer 1

It would have been nicer if the authors had included actual SIFT/SURF features instead of interest point counts. Surely the number of interest points alone will not give enough information about local patches as the actual features that are commonly used once these interest points are detected.

Response

We used two main predictive models: a CNN (which predicted complexity ratings from raw image data) and a linear regression (which predicted complexity ratings from 38 image statistics).

In response to the initial comments, we investigated the effect of mid-level features (SIFT and SURF) on image complexity. We extracted local keypoints from each image using the SIFT and SURF algorithms. We found between 7 and 4217 SIFT keypoints (mean 807) and between 12 and 4355 SURF keypoints (mean 1472).

How to characterise these keypoints? Since there were different numbers of keypoints for each image, a predictive model capable of accepting different input sizes was required. This ruled out our CNN model. The linear regression model is incapable of accepting keypoint descriptors as inputs as its inputs are scalar values.

We thus attempted to predict complexity from the keypoints themselves.

To do this, we attempted to characterise the keypoints by sampling 500 keypoints from each image, with replacement. This allowed us to analyse the character of the keypoints while controlling for their number (which is treated by the feature-based models). We assembled each image's keypoints into a vector of length $500 \times 128 = 64,000$ (SIFT) or $500 \times 64 = 32,000$ (SURF).

Image vectors were then used to train a support vector regression on a randomly selected 90% training set. We were not able to find parameters enabling the SVM to make effective predictions, so we do not report these results here.

However, a neural network was able to predict complexity values from sampled SIFT and SURF descriptors. We trained a 4-layer feedforward neural network with 38 inputs and {128, 64, 32, 1} units. ReLU activation was used throughout. We performed 10-fold cross-validation. The mean correlation with complexity ratings was $r = .45$, showing that SIFT and SURF descriptors can inform on perceived complexity, but are not the best predictors. Further work could use an RNN instead

of a feedforward neural network, allowing all keypoints from images with different numbers of keypoints to be taken into account.

These results are reported in sections 3.c and 3.e.

We have updated the manuscript to clarify these issues.

We would like to thank the reviewers once again for their detailed feedback. We believe that the revised paper has improved as a result of addressing all the points raised.

Sincerely,

Fintan Nagle
Nilli Lavie